# Clinical Manifestation, Management, and Outcomes in Patients with COVID-19 Vaccine-Induced Acute Encephalitis: Two Case Reports and a Literature Review

**DOI:** 10.3390/vaccines10081230

**Published:** 2022-07-31

**Authors:** Shiuan Shyu, Hua-Tung Fan, Shih-Ta Shang, Jenq-Shyong Chan, Wen-Fang Chiang, Chih-Chien Chiu, Ming-Hua Chen, Hann-Yen Shyu, Po-Jen Hsiao

**Affiliations:** 1Department of Chinese Medicine, College of Medicine, Chang Gung University, Taoyuan 333, Taiwan; susanis93171@gmail.com; 2Department of Internal Medicine, Taoyuan Armed Forces General Hospital, Taoyuan 325, Taiwan; talon4589@hotmail.com; 3Infectious Disease, Department of Internal Medicine, Taoyuan Armed Forces General Hospital, Taoyuan 325, Taiwan; stshang@aftygh.gov.tw (S.-T.S.); calebchiu.tw@gmail.com (C.-C.C.); 4Division of Nephrology, Department of Internal Medicine, Taoyuan Armed Forces General Hospital, Taoyuan 325, Taiwan; jschan0908@yahoo.com.tw (J.-S.C.); wfc96076@yahoo.com.tw (W.-F.C.); 5Division of Nephrology, Department of Internal Medicine, Tri-Service General Hospital, National Defense Medical Center, Taipei 114, Taiwan; 6Division of Neurology, Department of Internal Medicine, Taoyuan Armed Forces General Hospital, Taoyuan 325, Taiwan; son0715@aftygh.gov.tw; 7Department of Life Sciences, National Central University, Taoyuan 320, Taiwan; 8Department of Medicine, Fu-Jen Catholic University Hospital, College of Medicine, Fu Jen Catholic University, New Taipei City 242, Taiwan; 9Institute of Cellular and System Medicine, National Health Research Institutes, Miaoli County 350, Taiwan

**Keywords:** COVID-19 vaccine, mRNA-1273 vaccine, moderate, steroid, encephalitis

## Abstract

Introduction: Vaccination is one of the best strategies to control coronavirus disease 2019 (COVID-19), and multiple vaccines have been introduced. A variety of neurological adverse effects have been noted after the implementation of large-scale vaccination programs. Methods: We reported two rare cases of possible mRNA-1273 vaccine-induced acute encephalitis, including clinical manifestations, laboratory characteristics, and management. Results: The clinical manifestations might be related to hyperproduction of systemic and cerebrospinal fluid (CSF) cytokines. mRNA vaccines are comprised of nucleoside-modified severe acute respiratory syndrome coronavirus 2 (SARS-CoV-2) mRNA, which is translated into SARS-CoV-2 spike protein by the host’s ribosomes, activating the adaptive immune response. Exposed mRNA or vaccine components may also be detected as antigens, further resulting in aberrant proinflammatory cytokine cascades and activation of immune signaling pathways. Both patients exhibited significant clinical improvement after a course of steroid therapy. Conclusions: The use of COVID-19 vaccines to prevent and control SARS-CoV-2 infections and complications is the most practicable policy worldwide. However, inaccurate diagnosis or other diagnostic delays in cases of vaccine-induced acute encephalitis may have devastating and potentially life-threatening consequences for patients. Early diagnosis and timely treatment can result in a favorable prognosis.

## 1. Introduction

Vaccinating as many individuals as possible against coronavirus disease 2019 (COVID-19) is one of the most effective strategies for controlling the pandemic. In addition to complications like myocarditis and pericarditis, other neurological adverse effects have been reported following vaccination with messenger RNA (mRNA) vaccines, including the BNT162b2 vaccine (Comirnaty^®^ (New York, NY, USA); Pfizer-BioNTech (Mainz, Germany)) and mRNA-1273 vaccine (SPIKEVAX™; Moderna; Cambridge, MA, USA). We reported two rare cases of possible mRNA vaccine-induced acute encephalitis, both of whom had a favorable prognosis after steroid therapy. We speculate that the excessive innate immune response is due to cytokine storms triggered by vaccination. In certain individuals, vaccine components may be detected as antigens, triggering aberrant proinflammatory cytokine cascades and activation of immune signaling pathways, resulting in inflammatory symptoms, and secondary organ damage [1,2,3,4,5,6,7]. While vaccination provides substantial benefits and a means to eventually control the COVID-19 pandemic, clinicians should also be aware of the potential for vaccine-induced severe neurological complications.

## 2. Case Presentation

### 2.1. Case 1

A healthy 58-year-old woman was admitted due to acute delirium 7 days after receiving the mRNA-1273 vaccination (SPIKEVAX™). Prior to the recent vaccination, she had also received two doses of the Vaxzevria^®^ (ChAdOx1 nCov-19; AstraZeneca (Cambridge, UK)) vaccine 11 and 27 weeks before, without experiencing significant adverse effects. She had no history of neurological disorders. Physical examination revealed a low-grade fever (38 °C), cognitive deficits, left deviation of the head and eyeballs, and mild weakness of the right upper limb. Laboratory results, including complete blood cell counts, blood sugar levels, electrolyte levels, liver function tests, kidney function tests, and urinalysis, were normal (Table 1). A real-time reverse-transcription polymerase chain reaction (RT–PCR) for severe acute respiratory syndrome coronavirus 2 (SARS-CoV-2) was negative. Chest X-ray and brain computed tomography showed no obvious abnormalities.

A lumbar puncture was performed, and cerebrospinal fluid (CSF) in Case 1 was analyzed. Laboratory tests revealed lymphocyte-predominant pleocytosis (white blood cell (WBC) count: 40/µL, 59% lymphocytes), an elevated protein level of 82.9 (reference range: 15–45) mg/dL, an elevated CSF/serum albumin ratioof 19.7 (reference range: 5–8) × 10^−3^, a normal glucose level of 61.72 (reference range: 40–70) mg/dL, and a normal immunoglobulin G (IgG) index of 0.32 (reference range: 0.0–0.7). The patient was initially diagnosed with encephalitis based on the following clinical symptoms: (1) altered mental status lasting ≥24 h with no alternative cause, (2) documented fever within 72 h, (3) new onset of focal neurological findings, and (4) laboratory results (WBC count ≥ 5/µL) [8]. Intravenous empiric antibiotics and antiviral drugs, including ceftriaxone, vancomycin, and acyclovir, were initiated to treat acute encephalitis with an unknown cause. After 2 days of treatment, the patient’s symptoms persisted, without any improvement. CSF microbiological tests were negative for herpes simplex virus-1 (HSV-1), HSV-2, tuberculosis (TB), and bacterial culture; the venereal disease research laboratory (VDRL) test was also negative. Additionally, the patient’s influenza A and B viral nasal swab PCR tests, CSF cytological examination, and autoimmune encephalitis panel were negative (Table 2). Moreover, the patient’s blood tests for common pathogens and auto-antibodies (blood culture, virus serology, rheumatoid factor, antinuclear antibody (ANA), antithyroid peroxidase antibody, antimitochondrial, etc.) were all negative (Table 3). Brain magnetic resonance imaging (MRI) with contrast showed unremarkable findings. Finally, a diagnosis of COVID-19 vaccine-induced acute encephalitis was made. Dexamethasone (40 mg per day) was added on the 3rd day, and the patient exhibited a dramatic improvement on the next day. She regained normal cognitive function and displayed no further neurological impairment. We maintained treatment with intravenous steroids and gradually halved the dosage every 3 days. The patient was uneventfully discharged on the 13th day.

### 2.2. Case 2

A 21-year-old male was admitted to the Emergency Department due to coma approximately one week after receiving the mRNA-1273 (SPIKEVAX™) vaccination. The patient had no history of seizures, and the family history was unremarkable. RT–PCR results for SARS-CoV-2 were negative. Complete blood counts and electrolytes were normal (Table 4). Chest X-ray, brain computed tomography, and electrocardiography showed no obvious abnormalities. He experienced an episode of status epilepticus in the emergency department and was transferred to the intensive care unit (ICU) for further management. A lumbar puncture was performed. CSF analysis revealed no pleocytosis, an elevated protein level of 65.5 (normal range: 15–45) mg/dL, and an elevated microalbumin level of 37 (normal range: <6.5) pg/dL (Table 5). Although brain MRI with contrast was unremarkable, electroencephalography (Appendix A) revealed a continuous diffuse slowing in the theta and delta ranges, indicating moderate diffuse cerebral dysfunction (3rd hospital day). The cerebral perfusion scan with single-photon emission computed tomography (SPECT) indicated hypoperfusion in the right temporal region (Appendix A), which was compatible with the probable seizure origin. The patient was also diagnosed with encephalitis based on the following clinical symptoms: (1) altered mental status lasting ≥24 h with no alternative cause, (2) generalized seizures not fully attributable to a pre-existing seizure disorder, (3) abnormal electroencephalography results, and (4) abnormal neuroimaging of the brain parenchyma [8]. The test results for HSV, VDRL, TB, and other bacterial and fungal cultures of the CSF were all negative (Table 5). Similar to Case 1, the results of Case 2′s blood tests for common pathogens and auto-antibodies were also negative (Table 6). The autoimmune antibody tests for limbic encephalitis (anti-NMDR, anti-AMPAR1, anti-AMPAR2, anti-GABABR, anti-LGI1, anti-CASPR2) were also negative. A final diagnosis of COVID-19 vaccine-induced encephalitis complicated by seizures was made.

To prevent oxidative stress and maintain cellular homeostasis, controlling status epilepticus, intravenous levetiracetam and valproate sodium were administered. His seizures persisted in the ICU, and pulse corticosteroid therapy was administered on the 6th day of hospitalization with 1000 mg of intravenous methylprednisolone. We gradually halved the dosage every 3 days during the total 21-day hospital stay (14-day ICU stay). The patient’s clinical condition improved significantly after steroid administration. He was seizure-free during the rest of the hospital stay as well as at a 3-month outpatient department (OPD) follow-up.

## 3. Discussion

Given the absence of evidence of infection and the dramatic improvement after receiving corticosteroid treatments in both cases, we assumed that an immune-mediated mechanism was responsible for the presentation of acute encephalitis in both patients. In addition, both patients failed to meet the clinical diagnostic criteria for paraneoplastic or autoimmune encephalitis [9]. Therefore, we believe that the COVID-19 vaccine is the only possible cause of acute encephalitis in our patients, given the temporal proximity of receiving the COVID-19 vaccine and the lack of other risk factors for encephalitis.

A variety of postvaccination neurological complications have been reported since the introduction of the COVID-19 vaccines, but the underlying pathological mechanism remains unclear [1,2,3,4,10,11,12,13,14,15,16,17,18,19,20,21,22,23]. In addition to vaccines for COVID-19, postvaccination encephalitis has also been reported in association with several other vaccines, including those for measles, yellow fever, and smallpox [24]. mRNA vaccines consist of nucleoside-modified SARS-CoV-2 mRNA, which is translated into the SARS-CoV-2 spike protein by the host’s ribosomes, thus activating the adaptive immune response. However, exposed mRNA or vaccine components may be detected as antigens in certain individuals, triggering aberrant proinflammatory cytokine cascades and activation of immune signaling pathways [1,2,3,4,5,6]. These responses may result in elevated levels of circulating cytokines, inflammatory symptoms, and secondary organ damage. The underlying pathophysiology of cytokine-related neurotoxicity may resemble immune effector cell-associated neurotoxicity syndrome [25]. Furthermore, the spike protein alone can disrupt the blood–brain barrier (BBB), resulting in an increased BBB permeability, which may allow the overproduced inflammatory substances to enter the central nervous system. The elevation of the CSF/serum albumin ratio in both patients indicated impairment of the BBB, possibly due to disruption of cerebrovascular endothelial cells by the spike protein [26,27,28]. This brief report does not challenge the benefits of vaccination, but it does suggest caution and can guide management and provide prognosis for such patients. Larger epidemiological studies or meta-analyses are needed to understand the underlying mechanisms of postvaccination encephalitis. Presently, the benefits of COVID-19 vaccination outweigh any potential risks. The innate immune response of these two may explain this phenomenon, but further studies are needed to clarify the pathophysiology. We also reviewed the literature and compared clinical manifestations, management, and outcomes in patients with COVID-19 mRNA vaccine-induced acute encephalopathies (Table 7).

In summary, COVID-19 vaccinations generate antigens that may be recognized as potential pathogens by pattern-recognition receptors on resident stromal cells and circulating immune cells. Induction and transcription of specific genes may ensue, triggering the synthesis and release of pyrogenic cytokines, including interleukin [IL]-1, IL-6, tumor necrosis factor-alpha [TNF-α], and prostaglandin-E2, into the bloodstream, mimicking the response to natural infection. The cytokine-mediated inflammatory process is proposed to be the key pathophysiological mechanism underlying COVID-19 vaccine-related encephalitis [1,2,3,4,5,6,7].

## 4. Conclusions

COVID-19 vaccine-induced acute encephalitis is rare but may occur in clinical practice. This condition is characterized by activation of the immune response, triggering cytokine storm-mediated inflammation; misdiagnosis or delayed diagnosis may lead to fatal complications. Appropriate corticosteroid administration may be an effective treatment method in these patients [1,2,3,4,13,14,15,16,17].

## Figures and Tables

**Table 1 vaccines-10-01230-t001:** Results of blood biochemistry tests and complete blood cell counts (Case 1-upon admission).

Parameter	Result	Unit	Normal Range
BUN	15.2	mg/dL	7–25
Creatinine	0.68	mg/dL	F: 0.44–1.03; M: 0.64–1.27
eGFR	94.5	ml/min/1.732 m^2^	
Sodium	140.8	mmol/L	136–146
Potassium	4.1	mmol/L	3.5–5.1
Calcium	8.6	mg/dL	8.6–10.3
Chloride	105.9	mmol/L	101–109
GOT	17	mmol/L	Adult: ≤34
GPT	21	mmol/L	Adult: ≤36
Total bilirubin	0.87	mmol/L	0.3–1.2 (5 days-60 y)
Glucose	102.9	mg/dL	AC: 74–100 (≥18 y)PC: <140 (≥18 y)
White blood cell count	8.01	10^3^/µl	M: 3.9–10.6; F: 3.5–11
Red blood cell count	4.5	10^6^/µl	M: 4.5–5.9; F: 4.0–5.2
Hemoglobin	14.1	g/dL	M: 13.5–17.5; F: 12–16
Hematocrit	44.4	%	M: 41–53; F36–46
MCV	98.7	fl	80–100
MCH	31.3	pg	26–34
MCHC	31.8	g/dl	31–37
Platelet count	194	10^3^/mm	150–400

BUN: blood urea nitrogen; eGFR: estimated glomerular filtration rate; GOT: glutamyl oxaloacetic transaminase; GPT: glutamyl pyruvate transaminase; MCV: mean corpuscular volume; MCH: mean corpuscular hemoglobin; and MCHC: mean corpuscular hemoglobin concentration.

**Table 2 vaccines-10-01230-t002:** Results of the cerebrospinal fluid (CSF) examination (Case 1).

Parameter	Result	Unit	Normal Range
White blood cells	40	count/μL	0–5
Neutrophils	0	%	
Eosinophils	0	%	
Monocytes	41	%	
Lymphocytes	59	%	
Red blood cells	26	count/μL	0–5
pH	7.28		7.35–7.4
Total protein	82.9	mg/dL	15–45
LDH	16.1	U/L	<40
Chloride	126.5	mmol/L	118–132
Glucose	61.7	mg/dL	40–70
Albumin	55.1	mg/dL	10–30
CSF/serum albumin ratio (× 10^−3^)	19.7		5–8
IgG index	0.32		0–0.7
HSV 1 PCR	Not detected		
HSV 2 PCR	Not detected		
VDRL test	Negative		
CSF bacterial culture	No growth		
Gram stain	Negative		
Indian Ink	Not found		
TB PCR DNA	Negative		
Acid-Fast Stain	Not found		
TB culture	Negative		
Anti-NMDR	Negative		
Anti-AMPAR1	Negative		
Anti-AMPAR2	Negative		
Anti-GABABR	Negative		
Anti-LGI1	Negative		
Anti-CASPR2	Negative		

LDH: lactate dehydrogenase; HSV: herpes simplex virus; VDRL: venereal disease research laboratory; TB PCR: tuberculosis polymerase chain reaction; Anti-NMDR: anti-N-methyl-d-aspartate receptor; Anti-AMPAR: anti-α-amino-3-hydroxy-5-methyl-4 isoxazolepropionic acid receptor; Anti-GABABR: anti-r gamma-aminobutyric acid receptor; Anti-LGI1: anti-leucine-rich glioma inactivated-1; and Anti-CASPR2: anti-contactin-associated protein-like 2.

**Table 3 vaccines-10-01230-t003:** Results of encephalitis-related blood tests (Case 1).

Parameter	Result	Unit	Normal Range
HSV-1 IgG	Negative		
HSV-1 IgM	Negative		
HSV-2 IgG	Negative		
HSV-2 IgM	Negative		
CMV IgM	Negative		
EB-VCA IgM	Negative		
HBsAg	Nonreactive		
Anti-HCV	Nonreactive		
RSV screening test	Not detected		
Adenovirus Ag	Not detected		
Rotavirus Ag	Not detected		
PRP	Nonreactive		
TPPA	Nonreactive		
Cryptococcus Ag	Not detected		
Blood culture (2 sets)	No growth		
ANA	1:80		
Anti-dsDNA	1.4	U/mL	< 92.6
TSH	0.262	μL/U/mL	0.35–4.94
T3	0.66	ng/mL	0.64–1.52
Free T4	1.7	ng/mL	0.89–1.79
Anti-TPO Ab	<5	IU/mL	<5
Anti-thyroglobulin Ab	<15	IU/mL	<115
Anti-mitochondrial Ab	Negative		

HSV: herpes simplex virus; Ig: immunoglobulin; CMV: cytomegalovirus; EB-VCA: Epstein–Barr virus viral-capsid antigen; HBsAg: hepatitis B surface antigen; Anti-HCV: anti-hepatitis C virus; RSV: respiratory syncytial virus; PRP: rapid plasma reagin; TPPA: treponema pallidum particle agglutination assay; ANA: antinuclear antibodies; Anti-dsDNA: anti-double-stranded deoxyribonucleic acid; TSH: thyroid stimulating hormone; and Anti-TPO Ab: Anti-thyroid peroxidase antibody.

**Table 4 vaccines-10-01230-t004:** Results of biochemistry tests and complete blood counts (Case 2–upon admission).

Parameter	Result	Unit	Normal Range
BUN	12.8	mg/dL	7–25
Creatinine	0.92	mg/dL	F: 0.44–1.03; M: 0.64–1.27
eGFR	109.3	ml/min/1.732 m^2^	
Sodium	136.2	mmol/L	136–146
Potassium	4.2	mmol/L	3.5–5.1
GOT	20.9	mmol/L	Adult: ≤34
GPT	33.5	mmol/L	Adult: ≤36
CRP	2.68	ng/mL	<5
Glucose	103.0	mg/dL	AC: 74–100 (≥18 y); PC: <140 (≥18 y)
White blood cells	5.23	10^3^/µL	M: 3.9–10.6; F: 3.5–11
Red blood cells	5.38	10^6^/µL	M: 4.5–5.9; F: 4.0–5.2
Hemoglobin	15.1	g/dL	M: 13.5–17.5; F: 12–16
Hematocrit	45.8	%	M: 41–53; F: 36–46
MCV	85.1	fl	80–100
MCH	28.1	pg	26–34
MCHC	33.0	g/dL	31–37
Platelet count	241	10^3^/mm	150–400

BUN: blood urea nitrogen; eGFR: estimated glomerular filtration rate; GOT: glutamyl oxaloacetic transaminase; GPT: glutamyl pyruvate transaminase; MCV: mean corpuscular volume; MCH: mean corpuscular hemoglobin; and MCHC: mean corpuscular hemoglobin concentration.

**Table 5 vaccines-10-01230-t005:** Results of cerebrospinal fluid (CSF) examinations (Case 2).

Parameter	Result	Unit	Normal Range
White blood cells	<5	count/μL	0–5
Red blood cells	15	count/μL	0–5
pH	7.33		7.35–7.4
Total protein	65.5	mg/dL	15–45
LDH	16.0	U/L	<40
Chloride	125.7	mmol/L	118–132
Glucose	76.7	mg/dL	40–70
Albumin	37.0	mg/dL	10–30
CSF/serum albumin ratio (× 10^−3^)	8		5–8
IgG index	0.60		0–0.7
HSV 1 PCR	Not detected		
HSV 2 PCR	Not detected		
Influenza A	Not detected		
Influenza B	Not detected		
VDRL test	Negative		
CSF bacterial culture	No growth		
Gram stain	Negative		
Indian Ink	Not found		
TB PCR DNA	Negative		
Acid-Fast Stain	Not found		
TB culture	Negative		
Anti-NMDR	Negative		
Anti-AMPAR1	Negative		
Anti-AMPAR2	Negative		
Anti-GABABR	Negative		
Anti-LGI1	Negative		
Anti-CASPR2	Negative		

LDH: lactate dehydrogenase; HSV: herpes simplex virus; VDRL: venereal disease research laboratory; and TB PCR: tuberculosis polymerase chain reaction. Anti-NMDR: anti-N-methyl-d-aspartate receptor; Anti-AMPAR: anti-α-amino-3-hydroxy-5-methyl-4 isoxazolepropionic acid receptor; Anti-GABABR: anti-r gamma-aminobutyric acid receptor; Anti-LGI1: anti-leucine-rich glioma inactivated-1; and Anti-CASPR2: anti-contactin-associated protein-like 2.

**Table 6 vaccines-10-01230-t006:** Results of encephalitis-related blood tests (Case 2).

Parameter	Result	Unit	Normal Range
HSV-1 IgG	Negative		
HSV-1 IgM	Negative		
HSV-2 IgG	Negative		
HSV-2 IgM	Negative		
CMV IgM	Negative		
EB-VCA IgM	Negative		
*Varicella zoster* IgG	Negative		
Blood culture (2 sets)	Negative		
ANA	1:40 (negative)		
Anti-dsDNA	0.5	U/mL	<92.6
TSH	0.113	µl/U/mL	0.35–4.94
Free T4	1.43	ng/mL	0.89–1.79
Anti-Thyroglobulin Ab	15	IU/mL	<115

HSV: herpes simplex virus; Ig: immunoglobulin; CMV: cytomegalovirus; EB-VCA: Epstein–Barr virus viral-capsid antigen; ANA: antinuclear antibodies; Anti-dsDNA: anti-double-stranded deoxyribonucleic acid; TSH: thyroid stimulating hormone; and Anti-TPO Ab: Anti-thyroid peroxidase antibody.

**Table 7 vaccines-10-01230-t007:** Literature review of COVID-19 mRNA vaccine-induced acute encephalopathies.

Ref.	Diagnosis/ Clinical Feature	Vaccine/ Age and Sex/ Duration after Vaccination	Relevant Laboratory Data	Examinations/Images	Treatment
**[2]**	Aseptic meningitis - headache - fever (38 °C)	1st Comirnaty® BNT162b2 (BioNTech and Pfizer) 42 F/7 days	CSF exam: - protein level: normal - pleocytosis: 528/3 mm^3^ - glucose level: (-) - IgG index: normal Microbiological studies: negative	EEG: (-) CT: (-) MRI: (-)	Acyclovir and methylprednisolone 500 mg/day (3rd day) ⇨ Improved after steroid treatment; discharge on 5th day
**[3]**	Acute disseminated encephalomyelitis - unsteady gait - clumsiness of left arm	1st Comirnaty®; BNT162b2 (BioNTech and Pfizer) 56 F/14 days	CSF exam - protein level: normal - no pleocytosis - glucose level: normal Cytokine level - IL-10 CSF/serum ratio: 1.47 - IFN-gamma CSF/serum ratio: 3.66 - IL-6 CSF/serum ratio: 6.64 Microbiological studies: negative AE antibodies: negative Demyelinating disorder-related antigens: negative	EEG: normal CT: (-) MRI (FLAIR): hyperintensity involving the left cerebellar peduncle and supratentorial areas	Prednisone: 75 mg/day ⇨ Improvement 50 days after onset
**[10]**	Delirium - confusion - fluctuating attention - inversion of the sleep-wake cycle	1st Comirnaty®; BNT162b2 (BioNTech and Pfizer) 89 M/1 day	CSF exam: (-) Microbiological studies: (-)	EEG: (-) CT: (-) MRI: (-)	Quetiapine: 12.5 mg HS ⇨ Gradual improvement over the next 48 h
**[11]**	Movement disorder - restless movement - fever	2nd Comirnaty®; BNT162b2 (BioNTech and Pfizer) 36 F/12 h	CSF exam: (-) Microbiological studies: (-)	EEG: (-) CT: (-) MRI: (-)	Ibuprofen ⇨ Ceased spontaneously
**[4]**	Encephalopathy - disoriented - amnesic	1st SPIKEVAX™ mRNA-1273 (Moderna) 32 M/2 days	CSF exam: - protein level: elevated 0.76 gm/L - no pleocytosis - glucose level: normal Microbiological studies: negative AE antibodies: negative	EEG: slowed background activity CT: (-) MRI: normal	Ceftriaxone, acyclovir and methylprednisolone 1 g/day (6th day) ⇨ Dramatic improvement after steroid treatment
**[12]**	Encephalopathy - acute confusion - visual hallucination	1st SPIKEVAX™ mRNA-1273 (Moderna) 86 F/7 days	CSF exam: (-) Microbiological studies: negative	EEG: nonconvulsive focal status epilepticus CT: normal MRI: normal	Ceftriaxone, lorazepam (2nd day) and fosphenytoin (2nd day) ⇨ Significant improvement after anticonvulsant treatment
**[12]**	Encephalopathy - restlessness - cognitive deficits	1st SPIKEVAX™ mRNA-1273 (Moderna) 73 M/7 days	CSF exam: (-) Microbiological studies: negative	EEG: nonconvulsive status epilepticus CT: normal MRI: normal	Ceftriaxone, lorazepam, and levetiracetam ⇨ Significant improvement after anticonvulsant treatment
**[13]**	Encephalitis Sweet’s syndrome - confusion - orofacial movements - myoclonus - fever - generalized rash	1st SPIKEVAX™ mRNA-1273 (Moderna) 77 M/1 day	CSF exam: - protein level: elevated (124 m/dL) - pleocytosis (120 × 10^6^/L) - glucose level: normal Microbiological studies: negative AE antibodies: negative	EEG: generalized slow background in the theta range, with state changes and reactivity but no sleep features CT: normal MRI: normal	Vancomycin, ampicillin cefepime, ceftriaxone doxycycline, acyclovir and methylprednisolone 1 g/day ⇨ Returned to baseline before the 4th dose of methylprednisolone

(-): Laboratory data or examinations not mentioned in the references; CSF: cerebrospinal fluid; EEG: electroencephalogram; CT: computed tomography; MRI: magnetic resonance imaging; FLAIR: fluid attenuated inversion recovery; IL: interleukin; IFN: interferon; AE: autoimmune encephalitis; and HS: hora somni (taken at bedtime).

## Data Availability

The data underlying this article will be shared upon reasonable request by the corresponding author.

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
