# Peer review of "Clinical Manifestation, Management, and Outcomes in Patients with COVID-19 Vaccine-Induced Acute Encephalitis: Two Case Reports and a Literature Review"

_vaccines, 2022, doi:10.3390/vaccines10081230_

Round 1

Reviewer 1 Report

The article by Shyu, S. et al. describing their experience with two COVID-19 vaccine related encephalitis cases was notably improved according with reviewer´s suggestions. However, I would like to include some additional minor isssues:

-    Case nº 2, page 5, line 124. I think you could mention the antibodies used to detect limbic encephalitis.

-    Case nº2, page 5, lines 127-131. With the current description, it is still unclear the administration of corticoids. You mention 1000 mg every three days, followed by 500 mg every three days. You add that you tapered the dose during the 21 day hospital stay. The mentioned 1000 mg and 500 mg doses were applied once?

-    Discussion, page 7, line 164. I suggest “in an increased BBB permeability”.

-    Discussion, page 7, line 168. May be “inform” is not the adequate word here, or at least I am confused. May be “guide”, “direct” or “orient” are more in the context of the sentence. In addition, the final part of this sentence (“while anticipating further…”) is not clear for me; may be it can be included in the final paragraphs.

As a final note, I would like to congratulate the authors for their work, and apologize for the linguistic faults in my previous revision.

Author Response

Response to Reviewer 1

[General Comment]
The article by Shyu, S. et al. describing their experience with two COVID-19 vaccine related encephalitis cases was notably improved according with reviewer´s suggestions. However, I would like to include some additional minor isssues:

Author Reply: We sincerely appreciate the time and effort you spent reviewing this manuscript. We have revised the manuscript thoroughly according to your suggestions. Our responses to your comments are presented below.

[Comment 1]

Case nº 2, page 5, line 124. I think you could mention the antibodies used to detect limbic encephalitis.

Author Reply:

Thank you for your valuable comments. We revised the paragraphs as follows:

The autoimmune antibody tests for limbic encephalitis (Anti-NMDR, Anti-AMPAR1, Anti-AMPAR2, Anti-GABABR, Anti-LGI1, Anti-CASPR2) were also negative.

[Comment 2]

Case nº2, page 5, lines 127-131. With the current description, it is still unclear the administration of corticoids. You mention 1000 mg every three days, followed by 500 mg every three days. You add that you tapered the dose during the 21 day hospital stay. The mentioned 1000 mg and 500 mg doses were applied once?

Author Reply:

Thank you for your valuable comments. We apologized for the confusion we made.

We started pulse corticosteroid therapy with 1000 mg of intravenous methylprednisolone, and we halved the dosage every three days. Therefore, the mentioned 1000 mg and 500 mg doses were both applied for three days.

We revised the paragraphs as follows:

His seizures persisted in the ICU, and pulse corticosteroid therapy was administered on the 6th day of hospitalization with 1000 mg of intravenous methylprednisolone. We gradually halved the dosage every 3 days during the total 21-day hospital stay (14-day ICU stay).

[Comment 3]

Discussion, page 7, line 164. I suggest “in an increased BBB permeability”.

Author Reply:

Thank you for your valuable comments.

We revised the paragraphs as follows:

Furthermore, the spike protein alone can disrupt the blood-brain barrier (BBB), resulting in an increase BBB permeability, which allows the overproduced inflammatory substances to enter the central nervous system.

[Comment 4]

 Discussion, page 7, line 168. May be “inform” is not the adequate word here, or at least I am confused. May be “guide”, “direct” or “orient” are more in the context of the sentence. In addition, the final part of this sentence (“while anticipating further…”) is not clear for me; may be it can be included in the final paragraphs.

Author Reply:

Thank you for your valuable comments.

We have corrected the manuscript according to your suggestion. The revised sentence reads as follows:

This brief report does not challenge the benefits of vaccination, but it does suggest caution and can guide management and provide prognosis for such patients. Larger epidemiological studies or meta-analyses are needed to understand the underlying mechanisms of postvaccination encephalitis.

[Comment 5]

As a final note, I would like to congratulate the authors for their work, and apologize for the linguistic faults in my previous revision.

Author Reply:

Finally, we are deeply honored by the time and effort you put into reviewing this manuscript. While revising our manuscript based on your comments, we were motivated to pursue additional studies to enhance our understanding.

Reviewer 2 Report

Abstract Line 28: before "vaccine-induced" add "prpbably", "possibly"

Line 180-185: cite!

Author Response

Response to Reviewer 2

[Comment 1]

 Abstract Line 28: before "vaccine-induced" add "probably", "possibly"

Author Reply:

Thank you for your valuable comments. We revised the paragraphs as follows:

We report 2 rare cases of possible mRNA-1273 vaccine-induced acute encephalitis, including clinical manifestations, laboratory characteristics, and management.

[Comment 2]

Line 180-185: cite!

Author Reply:

Thank you for your valuable comments.

We revised the paragraphs as follows:

In summary, COVID-19 vaccinations generate antigens that may be recognized as potential pathogens by pattern-recognition receptors on resident stromal cells and circulating immune cells. Induction and transcription of specific genes may ensue, triggering the synthesis and release of pyrogenic cytokines, including interleukin [IL]-1, IL-6, tumor necrosis factor-alpha [TNF-α], and prostaglandin-E2, into the bloodstream, mimicking the response to natural infection. The cytokine-mediated inflammatory process is proposed to be the key pathophysiological mechanism underlying COVID-19 vaccine-related encephalitis [1-7].

Reference

  1. Baldelli, L., et al., Hyperacute reversible encephalopathy related to cytokine storm following COVID-19 vaccine. J Neuroimmunol, 2021. 358: p. 577661.
  2. Saito, K., et al., Aseptic meningitis after vaccination of the BNT162b2 mRNA COVID-19 vaccine. Neurol Sci, 2021. 42(11): p. 4433-4435.
  3. Vogrig, A., et al., Acute disseminated encephalomyelitis after SARS-CoV-2 vaccination. Clin Neurol Neurosurg, 2021. 208: p. 106839.
  4. Al-Mashdali, A.F., Y.M. Ata, and N. Sadik, Post-COVID-19 vaccine acute hyperactive encephalopathy with dramatic response to methylprednisolone: A case report. Ann Med Surg (Lond), 2021. 69: p. 102803.
  5. Ruiz, J.T., et al., Adjuvants- and vaccines-induced autoimmunity: animal models. Immunol Res, 2017. 65(1): p. 55-65.
  6. Salemi, S. and R. D'Amelio, Could autoimmunity be induced by vaccination? Int Rev Immunol, 2010. 29(3): p. 247-69.
  7. Chen, Y., et al., New-onset autoimmune phenomena post-COVID-19 vaccination. Immunology, 2022. 165(4): p. 386-401.

Finally, we are deeply honored by the time and effort you put into reviewing this manuscript. While revising our manuscript based on your comments, we were motivated to pursue additional studies to enhance our understanding.

Reviewer 3 Report

The authors attended to all the comments.

Only as a suggestion, lines 151 to 162 could be moved from discussion to introduction to improving the manuscript’s readability.

In the discussion section, the authors could emphasize that Comirnaty and Spikevax are the vaccines most associated with encephalopathies development.  

Author Response

Response to Reviewer 3

[General Comment]
The authors attended to all the comments.

Author Reply: We sincerely appreciate the time and effort you spent reviewing this manuscript. We have revised the manuscript thoroughly according to your suggestions. Our responses to your comments are presented below.

[Comment 1]

Lines 151 to 162 could be moved from discussion to introduction to improving the manuscript’s readability.

Author Reply:

Thank you for your valuable comments.

We have corrected the manuscript according to your suggestion. The revised sentence reads as follows:

Introduction

Vaccinating as many individuals as possible against coronavirus disease 2019 (COVID-19) is one of the most effective strategies for controlling the pandemic. In addition to complications like myocarditis and pericarditis, other neurological adverse effects have been reported following vaccination with messenger RNA (mRNA) vaccines, including the BNT162b2 vaccine (Comirnaty®; BioNTech and Pfizer) and mRNA-1273 vaccine (SPIKEVAX™; Moderna). We reported 2 rare cases of possible mRNA vaccine-induced acute encephalitis, both of whom had a favorable prognosis after steroid therapy. We speculate that the excessive innate immune response is due to cytokine storms triggered by vaccination. In certain individuals, vaccine components may be detected as antigens, triggering aberrant proinflammatory cytokine cascades and activation of immune signaling pathways, resulting in inflammatory symptoms, and secondary organ damage [1-4]. While vaccination provides substantial benefits and a means to eventually control the COVID-19 pandemic, clinicians should also be aware of the potential for vaccine-induced severe neurological complications.

[Comment 2]

In the discussion section, the authors could emphasize that Comirnaty and Spikevax are the vaccines most associated with encephalopathies development.

Author Reply:

Thank you for your valuable comments. We thank the reviewer for allowing us to explain more regarding this issue. Systematic review and meta-analysis of post-COVID vaccination adverse events indicated that viral vector vaccine and mRNA vaccine have a similar percentage of acute disseminated encephalomyelitis. However, the viral vector vaccine has a higher percentage of intracranial hemorrhage and acute transverse myelitis than the mRNA vaccine [1].We hope that we have addressed the issues raised by the reviewer.

  1. ElSawi, H.A. and A. Elborollosy, Immune-mediated adverse events post-COVID vaccination and types of vaccines: a systematic review and meta-analysis. Egypt J Intern Med, 2022. 34(1): p. 44.

Finally, we are deeply honored by the time and effort you put into reviewing this manuscript. While revising our manuscript based on your comments, we were motivated to pursue additional studies to enhance our understanding.

This manuscript is a resubmission of an earlier submission. The following is a list of the peer review reports and author responses from that submission.

Round 1

Reviewer 1 Report

The article by Shyu, S. et al. describing their experience with two COVID-19 vaccine related encephalitis cases seems well organized and scientifically relevant. The work is simple and the cases seem well described. However, I would like to include a some major issues:

-    In the presentation of both cases, I miss information about the rationale of the diagnoses. Why did you initiate the treatment with empiric antibiotics and antiviral drugs in case 1? Why did you choose that particular drugs? While in case nº 1 it seems logical the diagnosis of encephalitis, the provided information of case nº2 makes encephalitis a bit “fairish”, mainly supported by the seizures, which have been very rarely reported. Can you provide more data supporting the encephalitis diagnosis in the second patient?

-    When were the steroids applied in patient nº2? How much time did the patient stay in the ICU? As in the previous patient you have to describe the rationale of the diagnosis and treatment much better.

-    I assume that this patient did not have relevant CSF or blood tests in addition to the existing information in the text, but you should include something about this patient; I would like to see the SPECT, and may be the electroencephalography.

The text has some minor linguistic faults and some additional isssues, mentioned together as minor issues:

-    Abstract, page 1, line 39. I think “for patients” is more correct.

-    Introduction, page 1, line 46. I think you should specify “COVID-19 pandemic”.

-    Introduction, page 1, line 46. In addition to complications like (…).

-    Introduction, page 2, lines 48-49. You should include appropriate (reg) or (TM) symbols after the names of commercial companies. This point should be applied through all the manuscript.

-    Introduction, page 2, lines 50-51. The second part of the sentence is not fluently writte. I propose “encephalitis, both having favorable prognosis thanks to the steroid therapy”.

-    Introduction, page 2, line 51. When the COVID-19 pandemic the cytokine storm became famous to be triggered by the virus itself. You can apply here the cytokine storm also to the vaccines, but you should be much more clear, please. You can even include some references about this point in the introduction or in the discussion section (as this issue is mentioned in both sections, which should be also revised).

-    Case nº 1, page 2, line 59. The correct form is “weeks prior to”. I suggest change “prior” for other word like “before”.

-    Case nº 1, pages 2-3, tables 1-2. I guess that these tables are referred to patient nº1, but you should clarify this point in the title of both tables.

-    Case nº 2, page 88. I suggest a comma after “first seizure”.

-    Case nº2, page 4, line 78. To the prevention of oxidative stress and maintenance of cellular homeostasis.

-    Discussion, page 4, line 114. This sentence is difficult to read, may be you can remove “non-autoimmune antibody detection”.

-    Discussion, page 4, line 116. “Considered”.

-    Discussion, page 4, line 116-117. I don´t like this sentence, may be “mediated mechanism was responsible of the presentation as acute encephalitis in both patients”.

-    Discussion, page 4, line 130. Please, include a comma after  “permeability”.

-    Discussion, page 4, line 136. Please, include a comma after “vaccination”.

-    Discussion, page 5, lines 143-145. Could you make clearer this sentence? Various repeated words that make some confusion. The whole paragraph, actually, needs revision to improve comprehensibility.

Reviewer 2 Report

The topic of vaccine-induced adverse effects is clinically important.  Here authors report on two cases of encephalitis suggesting their relationship with preceding anti-COVID-19 vaccination.  Unfortunately, this hypothesis is highly speculative and not properly supported by clinical and laboratory findings. As multiple infectious agents can be responsible for encephalitis not detectable material of few diagnosed pathogens can not exclude of infectious encephalitis. Similarly, the undetectable autoantibodies does not determine the lack of autimmune encephalitis (the quick beneficial effect of steroids can rather support this diagnosis). 

As authors suggest the role of post-vacicination cytokine storm in presented patients they are recommended to perform such a testing in CSF samples (e.g. by Luminex technique) and compare the results with control samples collected from confirmed virus-induced encephalitis and non infectious neurological cases. Such a simple additional study could support the presented paper.

Reviewer 3 Report

The manuscript entitled “Clinical Manifestation, Management, and Outcome in Patients with COVID-19 Vaccine-Induced Acute Encephalitis: Two Case Reports and Literature Review” show relevant results regarding adverse side effects of Covid-19 vaccination.

Some points to take into account are enlisted below:

Although the title mentions that it encompasses a literature review, there are several relevant papers related to COVID-19 vaccination-associated neurological adverse side effects that are not included, such as: “Patone, M., Handunnetthi, L., Saatci, D. et al. Neurological complications after the first dose of COVID-19 vaccines and SARS-CoV-2 infection. Nat Med 27, 2144–2153 (2021)”, “Garg, R.K., Paliwal, V.K. Spectrum of neurological complications following COVID-19 vaccination. Neurol Sci 43, 3–40 (2022)”, “Takata J, Durkin SM, Wong S, Zandi MS, Swanton JK, Corrah TW. A case report of ChAdOx1 nCoV-19 vaccine-associated encephalitis. BMC Neurol. 2021 Dec 13;21(1):485.”, and “Vogrig A, Janes F, Gigli GL, Curcio F, Negro ID, D'Agostini S, Fabris M, Valente M. Acute disseminated encephalomyelitis after SARS-CoV-2 vaccination. Clin Neurol Neurosurg. 2021 Sep;208:106839.”

The second case report is not well described as the first one. The results of both should be included in the table.

If possible, include each patient's hemogram and blood biochemistry.

In the discussion section is important that the authors compare their results with the results of other studies that also reported Covid-19 vaccination-associated encephalitis, taking into account the vaccine type based on mRNA using adenoviral vector or liposomes. In addition, the author should discuss if there are other reports of the kinds of vaccines are associated with neurological adverse effects such as the reported by: “Alicino C, Infante MT, Gandoglia I, Miolo N, Mancardi GL, Zappettini S, Capello E, Orsi A, Tamburini T, Grandis M. Acute disseminated encephalomyelitis with severe neurological outcomes following virosomal seasonal influenza vaccine. Hum Vaccin Immunother. 2014;10(7):1969-73.”